

# Risk factors for osteoporosis in male patients with chronic obstructive pulmonary disease in Taiwan

Chu-Hsu Lin[1], Kai-Hua Chen[1,2], Chien-Min Chen[1,2], Chia-Hao Chang[3], Tung-Jung Huang[4,5] and Chia-Hung Lin[1]

[1] Department of Physical Medicine and Rehabilitation, Chang Gung Memorial Hospital, Chiayi, Puzi City, Chiayi County, Taiwan

[2] School of Medicine, Chang Gung University, Taoyuan City, Taiwan

[3] Department of Nursing, Chang Gung University of Science and Technology, Chiayi Campus, Puzi City, Chiayi County, Taiwan

[4] Division of Thoracic Medicine, Department of Internal Medicine, Chang Gung Memorial Hospital, Yunlin, Mailiao Township, Yunlin County, Taiwan

[5] Department of Respiratory Care, Chang Gung University of Science and Technology, Chiayi Campus, Puzi City, Chiayi County, Taiwan

Corresponding author
Tung-Jung Huang,
donaldhuang@cgmh.org.tw

## ABSTRACT

**Objective**. To investigate the risk factors for osteoporosis in male Taiwanese patients with chronic obstructive pulmonary disease (COPD).

**Methods**. This cross-sectional study evaluated male COPD outpatients and age-matched male subjects at a regional teaching hospital. The following data were obtained and analyzed: bone mineral density of the lumbar spine and hip on dual-energy X-ray absorptiometry, demographic characteristics, questionnaire interview results, pulmonary function test results, chest posterior–anterior radiographic findings, and biochemical and high-sensitivity C-reactive protein (hs-CRP) levels.

**Results**. Fifty-nine male COPD patients and 36 age-matched male subjects were enrolled. COPD patients had lower body mass index (BMI) (23.6 ± 4.1 vs. 25.2 ± 3.0 kg/m$^2$) and higher total prevalence for osteoporosis and osteopenia than controls. Among COPD patients, patients with osteoporosis had lower BMI, body weight, waist circumference, and triglyceride level but higher hs-CRP level, and tended to have lower creatinine level. Binary logistic regression analysis for factors including age, BMI, creatinine, hs-CRP, smoking, steroid use, and forced expiratory volume in one second (FEV1) revealed that an hs-CRP level ≥5 and decreased creatinine level were independent risk factors for osteoporosis in COPD patients. Lower BMI tended to be associated with osteoporosis development, although it did not reach statistical significance, and hs-CRP was associated with COPD severity and steroid use history.

**Conclusion**. The total prevalence of osteoporosis and osteopenia in male Taiwanese COPD patients is higher than that in age-matched male subjects and systemic inflammation is an independent risk factors for osteoporosis. Low creatinine level in COPD patients should raise the suspicion of sarcopenia and associated increased risk of osteoporosis.

## INTRODUCTION

Chronic obstructive pulmonary disease (COPD), a major global public health issue burdening health-care systems, is characterized by persistent and usually progressive airflow limitation resulting from chronic inflammation of the airways and lungs (*Vestbo et al., 2013*). The incidence and prevalence of COPD are much greater in men than in women worldwide (*Afonso et al., 2011*). Currently, it is regarded as a heterogeneous disease with many systemic manifestations and comorbidities, such as ischemic heart disease, heart failure, anemia, diabetes, skeletal muscle wasting, osteoporosis, and osteoporotic fracture (*Barnes & Celli, 2009*; *Lee et al., 2016*). It is also predicted to become the third leading cause of death in the world by 2020 (*Murray & Lopez, 1997*).

Osteoporosis, an important comorbidity of COPD, increases the risk of fracture, with resultant pain, functional limitation, and increased mortality (*Sambrook & Cooper, 2006*). Several studies showed that COPD patients have a high prevalence of osteoporosis or low bone mineral density (BMD), although estimates of prevalence are often underestimated and varied, depending on the study methodology and case enrollment criteria (*Rittayamai, Chuaychoo & Sriwijitkamol, 2012*; *Romme et al., 2013*). A recent systematic review revealed mean prevalences for osteoporosis and osteopenia in COPD patients as approximately 35.1% (9–69%) and 38.4% (27–67%), respectively (*Graat-Verboom et al., 2009*). Therefore, early detection, prevention, and treatment of osteoporosis in COPD patients is important.

The cause of osteoporosis in COPD patients is complex, and various risk factors are likely related to the pathogenesis, such as old age (*Kanis et al., 2008*); tobacco smoking (*Afonso et al., 2011*); female sex (*Ekstrom, Jogreus & Strom, 2012*); lower body mass index (BMI) (*Graat-Verboom et al., 2009*); physical inactivity (*Silva et al., 2011*); steroid use (*Loke, Cavallazzi & Singh, 2011*); lower fat-free mass (*Graat-Verboom et al., 2009*); and systemic inflammation markers such as C-reactive protein (CRP), tumor necrosis factor-alpha (TNF-$\alpha$), and interleukin-6 (IL-6) (*Liang & Feng, 2012*; *Rittayamai, Chuaychoo & Sriwijitkamol, 2012*). However, cumulative results of these studies are inconclusive, and the definite pathogenic mechanism remains unclear. To our knowledge, data regarding risk factors in Taiwan are limited, with no current research regarding the association between inflammation markers or systemic inflammation and osteoporosis development (*Chen et al., 2015*; *Lee et al., 2016*; *Lin et al., 2015*; *Liu et al., 2015*). Therefore, this study primarily aimed to investigate risk factors for osteoporosis, particularly whether increased CRP is a predictor for osteoporosis development in male Taiwanese COPD patients since it is a common and available test in the hospital.

## MATERIALS AND METHODS

### Subjects and study design

This cross-sectional study was a sub-program of community elderly medical research program. It enrolled male COPD patients aged $\geq$45 years from a chest outpatient clinic in a regional teaching hospital in the west Chiayi County, Taiwan. Patients were clinically stable without recent acute exacerbation in the previous 3 months. Age-matched male subjects were recruited from participants in health examinations at local public health

centers in the west Chiayi County. Subjects who were willing to undergo anthropometric measurements, questionnaire interviews, blood tests, chest radiography, spirometry, and BMD examination were included. Subjects with rheumatic disease, bronchial asthma, or other structural lung diseases (including lung cancer, bronchiectasis, and lung fibrosis) revealed by past medical history or chest radiography were excluded. Written informed consent was obtained from all participants, and the study was approved by the local Institutional Review Board of Chiayi Chang Gung Memorial Hospital (IRB number: 96-0495B).

## Anthropometric measurements

Body weight, height, and waist and hip circumferences were measured with participants wearing light clothing without shoes. BMI was calculated.

## Questionnaire interview and medical records review

All participants were interviewed using a comprehensive questionnaire. Age, medical history, dietary habits, and lifestyle factors including tobacco smoking, milk or calcium supplement consumption, exercise, and employment as a manual labor were recorded. If applicable, further information was collected on the amount, frequency, and duration of certain lifestyle habits. Habitual milk consumption was defined as drinking ≥7 glasses per week for ≥1 year. Subjects with habitual milk or calcium supplement consumption were considered to have a high calcium intake. Habitual exercise was defined as exercising ≥3.5 hours/week for ≥1 year. Subjects who were habitual exercisers or manual labors were defined as those with high physical activity. Past medical history (including hypertension, diabetes mellitus, dyslipidemia, liver disease, and renal disease) was recorded. If present, liver or renal disease was further defined.

Steroid use or history of acute exacerbation of COPD in the previous 3 years was determined from the medical records by a physician. Steroid use was classified as (1) oral for >6 months regardless of continuous or intermittent use, (2) inhaled for >6 months regardless of continuous or intermittent use, (3) occasional (either oral or inhaled for <6 months), or (4) not used (See Table 1).

## Blood tests

Biochemical parameters were determined for serum blood urea nitrogen, creatinine, total cholesterol, low-density lipoprotein cholesterol, high-density lipoprotein cholesterol, triglyceride, glycohemoglobin (HbA1c), uric acid, homocysteine, vitamin B12, folate, and high-sensitivity CRP (hs-CRP). Subjects with a CRP level above the normal reference value in the hospital (≥5 mg/dL) were classified as the group of increased hs-CRP.

## Lung examination

All subjects underwent posterior–anterior chest radiography and spirometry. Post-bronchodilator spirometry examination was performed by trained personnel using a KoKo spirometer (Pulmonary Data Services, Inc., Louisville, KY, USA). The following parameters were recorded: forced expiratory volume in 1 s (FEV1), FEV1 % of predicted value (FEV1% predicted), forced vital capacity (FVC), FVC % of predicted value (FVC%

**Table 1 Definition of some lifestyle habit or medical condition variables.**

| Variables | Habits/conditions | Frequency | Duration | Source |
|---|---|---|---|---|
| High calcium intake (milk or calcium supplement) | Habitual milk consumption | ≥7 glasses/week | ≥1 year | Questionnaire |
| | Calcium supplement consumption | Frequent use | ≥1 year | Questionnaire |
| High physical activity (habitual exercise or manual labor) | Habitual exercise | ≥3.5 hours/week | ≥1 year | Questionnaire |
| | Manual labor | As a job | ≥1 year | Questionnaire |
| COPD with acute exacerbation | Yes | Any acute exacerbation | During previous 3 years | Medical records |
| | No | No exacerbation | | |
| Steroid use | Not used | No steroid use history[a] | | |
| | Occasional | Cumulative steroid use <6 months | | |
| | Inhaled | Inhaled steroid >6 months in previous 3 years regardless of continuously or intermittently | During previous 3 years | Medical records |
| | Oral | Oral steroid use >6 months in previous 3 years regardless of continuously or intermittently | | |

**Notes.**

Abbreviations: COPD, Chronic Obstructive Pulmonary Disease.

[a]Some people did not have medical records for more than 3 years.

predicted), FEV1/FVC ratio, and forced expiratory flow 25–75% (FEF 25–75%). Chest radiography and spirometry data were interpreted by a chest physician. The diagnosis and classification of the severity (stages I–IV) of COPD were based on the diagnostic criteria of the Global Initiative for Chronic Obstructive Lung Disease (GOLD) guidelines (*Johannessen et al., 2013*).

## Bone mineral density examination

BMD measurements at the first to fourth lumbar vertebrae (L1–L4) and left hip (right hip if left hip BMD unreadable because of conditions such as a fracture) were obtained using dual-energy X-ray absorptiometry (Delphi A, QDR series; Hologic, Bedford, MA, USA).

According to the diagnostic criteria proposed by a World Health Organization working party (*Melton et al., 1993*), osteoporosis and osteopenia were defined as a BMD $T$-score at either the lumbar spine or the hip of $<-2.5$ and between $-1$ and $-2.5$, respectively.

## Data analysis

Continuous variables are presented as the mean $\pm$ standard deviation and categorical variables as the frequency and group percentage. Continuous and categorical variables were compared among groups using the independent sample $t$-test and Pearson's chi-square test or Fisher's exact test, respectively. Binary logistic regression analyses were used for multivariate analysis to assess which background variables were predictive of osteoporosis, and odds ratios were calculated with a 95% confidence interval (CI). A $P$-value of <0.05 was considered significant for all tests. Statistical analysis was performed with SPSS software, version 18 (SPSS, Inc., Chicago, IL, USA).

**Table 2  Clinical characteristics of COPD patients and control subjects.**

| Variables | Control group | | COPD group | | P-value |
|---|---|---|---|---|---|
| | n | Mean ± SD, percentage | n | Mean ± SD, percentage | |
| Age (years) | 36 | 71.1 ± 5.9 | 59 | 71.3 ± 7.4 | 0.846[a] |
| Body height (cm) | 36 | 161.4 ± 5.3 | 59 | 164.2 ± 6.1 | **0.028**[a] |
| Body weight (kg) | 36 | 65.8 ± 8.4 | 59 | 63.6 ± 11.8 | 0.340[a] |
| BMI (kg/m$^2$) | 36 | 25.2 ± 3.0 | 59 | 23.6 ± 4.1 | **0.035**[a] |
| L spine BMD (g/cm$^2$) | 35 | 1.03 ± 0.21 | 58 | 0.94 ± 0.19 | 0.054[a] |
| Hip BMD (g/cm$^2$) | 36 | 0.92 ± 0.13 | 57 | 0.84 ± 0.14S | **0.011**[a] |
| Smoking | 36 | | 59 | | **0.001**[b] |
| No smoking | | 69.4% (25/36) | | 28.8% (17/59) | |
| Quit | | 13.9% (5/36) | | 39.0% (23/59) | |
| Still smoking | | 16.7% (6/36) | | 32.2% (19/59) | |
| High physical activity | 36 | 63.9% (23/36) | 59 | 52.5% (31/59) | 0.279[b] |
| Comorbidity | | | | | |
| Hypertension | 36 | 38.9% (14/36) | 59 | 33.9% (20/59) | 0.623[b] |
| Diabetes mellitus | 36 | 2.8% (1/36) | 59 | 5.1% (3/59) | 1.000[c] |
| Dyslipidemia | 36 | 11.1% (4/36) | 59 | 8.5% (5/59) | 0.726[c] |
| Chronic kidney disease | 36 | 2.8% (1/36) | 59 | 1.7% (1/59) | 1.000[c] |
| Chronic hepatitis | 36 | 2.8% (1/36) | 58 | 1.7% (1/58) | 1.000[c] |

**Notes.**

Abbreviations: BMD, bone mineral density; BMI, body mass index; SD, standard deviation.

[a] P-value by independent sample t-test.

[b] P-value by Chi-square test.

[c] P-value by Fisher's exact test.

When the p-value of analysis reached statistical significance (i.e., <0.05), we showed it in bold.

## RESULTS

From May 2009 to August 2011, 59 male COPD patients and 36 age-matched male subjects were enrolled. The data of FEV1/FVC was all above 70% in control group, on the other hand, it was all below 70% in COPD patient and the mean FEV1 of COPD patients was 1.26 ± 0.47 L (51.0% ± 18.0% predicted). The mean age of the two groups was similar, but BMI were lower in COPD patients. More people in COPD group had history of cigarette smoking. Hip BMD was lower in COPD group. Lumbar spine BMD also tended to be lower in COPD group, although it did not reach statistical significance (Table 2). Among COPD patients, 17 (28.8%) exhibited osteoporosis and 25 (42.4%) osteopenia compared with 16.7% and 33.3%, respectively, in controls. The total prevalence of osteoporosis and osteopenia in COPD patients was significant higher than that in controls by the Person's Chi-square test (P = 0.038). However, in multivariate binary logistic regression analysis showed that low BMI was the independent risk factors of increased prevalence of osteoporosis and osteopenia and the variable COPD (COPD patients versus controls) lost its statistical significance (Table 3).

General clinical characteristics and laboratory findings among COPD patients with and without osteoporosis are shown in Table 4. Body weight, BMI, waist circumference, and triglyceride level were significantly lower in the osteoporosis group. The creatinine level

**Table 3** Binary logistic regression for multivariate analysis for the risk of increased prevalence of osteoporosis and osteopenia in COPD and healthy control subjects.

| Variables | B | SE | Odds ratio | 95% CI | | P-value |
|---|---|---|---|---|---|---|
| | | | | Lower | Upper | |
| Age (years) | −0.003 | 0.038 | 1.00 | 0.93 | 1.07 | 0.938 |
| BMI (kg/m$^2$) | −0.229 | 0.080 | 0.80 | 0.68 | 0.93 | **0.004** |
| COPD | 0.631 | 0.506 | 1.88 | 0.70 | 5.07 | 0.213 |
| hs-CRP ≥ 5 (mg/L) | 1.724 | 1.198 | 5.61 | 0.54 | 58.71 | 0.150 |
| Creatinine | −0.767 | 1.017 | 0.46 | 0.006 | 3.41 | 0.451 |
| Still smoking | −0.177 | 0.613 | 0.84 | 0.25 | 2.79 | 0.838 |
| Milk/Calcium supplement | 0.062 | 0.507 | 1.06 | 0.39 | 2.88 | 0.902 |
| Hight physical acitivity | −0.187 | 0.521 | 0.83 | 0.30 | 2.30 | 0.719 |
| Constant | 6.789 | 3.712 | 887.80 | | | 0.067 |

**Notes.**
Abbreviations: BMI, body mass index; COPD, chronic obstructive pulmonary disease; hs-CRP, hypersensitive C-reactive protein; FEV1, forced expiratory volume in one second; CI, confidence interval.
When the *p*-value of analysis reached statistical significance (i.e., <0.05), we showed it in bold.

tended to be lower in the osteoporosis group, although the statistical significance was marginal ($P = 0.050$). Patients with increased hs-CRP (≥5 mg/L) showed a high risk of osteoporosis compared with those with low hs-CRP although the mean value of hs-CRP in osteoporosis group was not significantly higher than that in non-osteoporosis group probably due to high standard deviation of the value.

Hs-CRP showed strong associations with COPD severity and pulmonary function parameters (including COPD GOLD stage, FEV1, FEV1% predicted, FVC, FVC% predicted, FEV1/FVC, and FEF 25–75%) and steroid use. The high hs-CRP group tended to have lower BMI, although there were no significant differences in age, smoking habits, and history of acute exacerbation (Table 5).

Factors related to COPD between the osteoporosis and non-osteoporosis groups are compared in Table 6. There was no significant association between osteoporosis and other variables (age, COPD severity, pulmonary function test parameters, calcium intake, physical activity, cigarette smoking, frequent exacerbation of COPD, and corticosteroid use).

Multivariate analysis using binary logistic regression including age, BMI, creatinine, and increased hs-CRP (category), still smoking (category), oral or inhaled steroid use for more than 6 months (category), and FEV1 showed that an hs-CRP level ≥5 and decreased creatinine level were independent risk factors for osteoporosis in COPD patients, with odds ratios of 58.90 (95% CI [2.09–1548.13]) and 0.01 (95% CI [0.00–0.67]), respectively. BMI tended to be negatively associated with osteoporosis development, although statistically insignificant (Table 7). Because body weight, waist circumference, and triglyceride level were strongly associated with BMI, and all of the other respiratory parameters were strongly associated with FEV1 by the Pearson correlation test, these factors were not included in the regression analysis.

**Table 4** Clinical characteristics and laboratory test of COPD patients with and without osteoporosis.

| Variables | Non-osteoporosis group | | Osteoporosis group | | P-value |
|---|---|---|---|---|---|
| | n | Mean ± SD, percentage | n | Mean ± SD, percentage | |
| Age (years) | 42 | 70.8 ± 7.2 | 17 | 72.8 ± 8.2 | 0.354[a] |
| L spine BMD (g/cm$^2$) | 41 | 1.02 ± 0.16 | 17 | 0.76 ± 0.10 | **<0.001**[a] |
| Hip BMD (g/cm$^2$) | 41 | 0.89 ± 0.12 | 16 | 0.72 ± 0.11 | **<0.001**[a] |
| Anthropometric data | | | | | |
| Body height (cm) | 42 | 164.0 ± 6.3 | 17 | 164.7 ± 6.0 | 0.664[a] |
| Body weight (kg) | 42 | 65.7 ± 11.8 | 17 | 58.4 ± 10.5 | **0.030**[a] |
| BMI (kg/m$^2$) | 42 | 24.4 ± 4.1 | 17 | 21.4 ± 3.2 | **0.009**[a] |
| Waist circumflex (cm) | 39 | 91.1 ± 11.7 | 16 | 83.2 ± 10.4 | **0.024**[a] |
| Hip circumflex (cm) | 38 | 94.1 ± 7.1 | 16 | 91.3 ± 6.1 | 0.174[a] |
| Laboratory test | | | | | |
| Bun (mg/dL) | 37 | 16.9 ± 5.1 | 14 | 15.7 ± 6.0 | 0.497[a] |
| Creatinine (mg/dL) | 39 | 1.2 ± 0.3 | 14 | 1.0 ± 0.2 | **0.050**[a] |
| Cholesterol (mg/dL) | 38 | 188.8 ± 35.1 | 14 | 184.0 ± 29.0 | 0.651[a] |
| HDL (mg/dL) | 38 | 52.6 ± 14.0 | 14 | 53.9 ± 11.1 | 0.765[a] |
| LDL (mg/dL) | 38 | 120.8 ± 30.6 | 14 | 119.6 ± 31.3 | 0.901[a] |
| Triglyceride (mg/dL) | 38 | 112.9 ± 61.6 | 14 | 79.9 ± 18.9 | **0.005**[a] |
| HbA1c (%) | 37 | 5.9 ± 0.9 | 14 | 5.7 ± 0.3 | 0.372[a] |
| Uric acid (mg/dL) | 38 | 6.8 ± 1.6 | 14 | 6.3 ± 1.3 | 0.311[a] |
| Homocysteine (umol/L) | 42 | 17.5 ± 21.7 | 17 | 12.6 ± 3.3 | 0.358[a] |
| Vitamin B12 (pg/mL) | 42 | 628.6 ± 287.1 | 17 | 713.8 ± 413.8 | 0.370[a] |
| Folate (ng/mL) | 40 | 11.5 ± 7.3 | 17 | 9.8 ± 3.9 | 0.363[a] |
| hs-CRP (mg/L); (≥5 (mg/L)) | 41 | 2.60 ± 3.58; 7.3% (3/41) | 16 | 22.63 ± 47.06; 31.3% (5/16) | 0.109[a]; **0.032**[c] |
| Milk/Calcium supplement | 42 | 40.5% (17/42) | 17 | 35.3% (6/17) | 0.712[b] |
| High physical activity | 42 | 47.6% (20/42) | 17 | 64.7% (11/17) | 0.234[b] |
| Comorbidity | | | | | |
| Hypertension | 42 | 35.7% (15/42) | 17 | 29.4% (5/17) | 0.643[b] |
| Diabetes mellitus | 42 | 7.1% (3/42) | 17 | 0 (0/17) | 0.550[c] |
| Dyslipidemia | 42 | 33.3% (4/42) | 17 | 5.9% (1/17) | 1.000[c] |
| Chronic kidney disease | 42 | 2.4% (1/42) | 17 | 0 (0/17) | 1.000[c] |
| Chronic hepatitis | 41 | 2.4% (1/41) | 17 | 0 (0/17) | 1.000[c] |

**Notes.**

Abbreviations: BMD, bone mineral density; BMI, body mass index; HDL, high density lipoprotein; LDL, low density lipoprotein; SD, standard deviation.

[a]P-value by independent sample *t*-test.

[b]P-value by Chi-square test.

[c]P-value by Fisher's exact test.

When the *p*-value of analysis reached statistical significance (i.e., <0.05), we showed it in bold.

## DISCUSSION

The present study revealed that the prevalences of osteoporosis and osteopenia in COPD patients at a community hospital in Taiwan were 28.8% and 42.4%, respectively, and the total prevalence of them were higher than those in age-matched healthy controls, which was mainly contributed by low BMI in COPD patients. This is in agreement with previous studies showing that the risk of osteoporosis was greater in COPD patients than in healthy

**Table 5 Comparison of COPD patients with high and low hypersensitive C-reactive protein level.**

| Variables | hs-CRP < 5 (mg/L) | | hs-CRP ≥ 5 (mg/L) | | P-value |
|---|---|---|---|---|---|
| | N | Mean ± SD, percentage | n | Mean ± SD, percentage | |
| Age (years) | 49 | 71.0 ± 7.4 | 8 | 71.0 ± 7.3 | 0.988[a] |
| BMI | 49 | 24.0 ± 4.1 | 8 | 21.4 ± 3.5 | 0.092[a] |
| COPD GOLD stage | 49 | | 8 | | **0.023[b]** |
|   Stage I | | 14.3% (7/49) | | 0 (0/8) | |
|   Stage II | | 20.8% (20/49) | | 25% (2/8) | |
|   Stage III | | 42.9% (21/49) | | 37.5% (3/8) | |
|   Stage IV | | 2.0% (1/49) | | 37.5% (3/8) | |
| Pulmonary function test | | | | | |
|   FEV1 (L) | 49 | 1.36 ± 0.44 | 8 | 0.78 ± 0.33 | **0.001[a]** |
|   FEV1% predicted (%) | 49 | 54.4 ± 17.4 | 8 | 34.1 ± 14.4 | **0.003[a]** |
|   FVC (L) | 49 | 2.16 ± 0.60 | 8 | 1.40 ± 0.41 | **0.001[a]** |
|   FVC% predicted (%) | 49 | 67.8 ± 16.7 | 8 | 50.1 ± 20.8 | **0.010[a]** |
|   FEV1/FVC (%) | 49 | 62.4 ± 8.8 | 8 | 54.3 ± 10.3 | **0.022[a]** |
|   FEV25-75 (%) | 49 | 32.7 ± 14.2 | 8 | 17.9 ± 7.7 | **0.000[a]** |
| Smoking | 49 | | 8 | | 0.165[b] |
|   No smoking | | 30.6% (15/49) | | 12.5% (1/8) | |
|   Quit | | 34.7% (17/49) | | 75.0% (6/8) | |
|   Still smoking | | 34.7% (17/49) | | 12.5% (1/8) | |
| Acute exacerbation | 43 | 30.2% (13/43) | 8 | 62.5% (5/8) | 0.112[c] |
| Steroid used | 49 | | 8 | | **0.005[b]** |
|   No steroid used | | 44.9% (22/49) | | 0 (0/8) | |
|   Occasional | | 12.2% (6/49) | | 12.5% (1/8) | |
|   Inhaled | | 34.7% (17/49) | | 37.5% (3/8) | |
|   Systemic | | 8.2% (4/49) | | 50% (4/8) | |

**Notes.**

Abbreviations: hs-CRP, hypersensitive C-reactive protein; GOLD, global Initiative for Chronic Obstructive lung Disease; FEV1, forced expiratory volume in one second; FEV1% predicted, forced expiratory volume in one second of predicted value; FVC, forced vital capacity; FVC% predicted, forced vital capacity of predicted value; FEF 25–75, forced expiratory flow 25–75%.

[a] P-value by independent sample t-test.
[b] P-value by Fisher's exact test.
[c] P-value by Chi-square test.
When the p-value of analysis reached statistical significance (i.e., <0.05), we showed it in bold.

subjects (*Graat-Verboom et al., 2009*; *Rittayamai, Chuaychoo & Sriwijitkamol, 2012*; *Schnell et al., 2012*). However, the prevalence of osteoporosis in our study was lower than the 40% reported in a recent study also conducted in Chiayi, Taiwan (*Lin et al., 2015*). This may be explained by differences in subject enrollment and osteoporosis definition. That study enrolled male and female COPD patients and defined osteoporosis as a BMD $T$-score $<-2.5$ or the presence of thoracolumbar vertebral compression fracture on radiography.

This study also revealed that increased hs-CRP, decreased creatinine, and decreased BMI and its related parameters such as body weight, waist circumference, and triglyceride level were associated with osteoporosis in COPD patients in univariate analysis. Multivariate analysis with binary logistic regression revealed increased hs-CRP, and decreased creatinine were independent risk factors for osteoporosis development in COPD patients, which

**Table 6  COPD-related factors in patients with and without osteoporosis.**

| Variables | Non-osteoporosis group | | Osteoporosis group | | P-value |
|---|---|---|---|---|---|
| | N | Mean ± SD, percentage | n | Mean ± SD, percentage | |
| COPD GOLD stage | 42 | | 17 | | 0.271[a] |
| Stage I | | 16.7% (7/42) | | 0 (0/17) | |
| Stage II | | 33.3% (14/42) | | 52.9% (9/17) | |
| Stage III | | 42.9% (18/42) | | 41.2% (7/17) | |
| Stage IV | | 7.1% (3/42) | | 5.9% (1/17) | |
| Pulmonary function test | | | | | |
| FEV1 (L) | 42 | 1.29 ± 0.49 | 17 | 1.18 ± 0.44 | 0.428[b] |
| FEV1% predicted (%) | 42 | 53.1 ± 19.4 | 17 | 47.8 ± 13.8 | 0.313[b] |
| FVC (L) | 42 | 2.06 ± 0.67 | 17 | 1.97 ± 0.53 | 0.619[b] |
| FVC% predicted (%) | 42 | 66.4 ± 19.0 | 17 | 63.5 ± 15.7 | 0.589[b] |
| FEV1/FVC (%) | 42 | 62.0 ± 9.2 | 17 | 58.8 ± 9.5 | 0.233[b] |
| FEF 25–75 (%) | 42 | 31.6 ± 15.1 | 17 | 27.3 ± 11.5 | 0.294[b] |
| Smoking | 42 | | 17 | | 0.956[c] |
| No smoking | | 28.6 % (12/42) | | 29.4% (5/17) | |
| Quit | | 38.1% (16/42) | | 41.2% (7/17) | |
| Still smoking | | 33.3% (14/42) | | 29.4% (5/17) | |
| Acute exacerbation | 37 | 32.4% (12/37) | 16 | 43.8% (7/16) | 0.430[c] |
| Steroid used | 42 | | 17 | | 0.946[a] |
| Oral | | 11.9% (5/42) | | 17.6% (3/17) | |
| Inhaled | | 38.1% (16/42) | | 35.3% (6/17) | |
| Occasional | | 11.9% (5/42) | | 11.8% (2/17) | |
| No steroid used | | 38.1% (16/42) | | 35.3% (6/17) | |

**Notes.**

Abbreviations: GOLD, global Initiative for Chronic Obstructive lung Disease; FEV1, forced expiratory volume in one second; FEV1% predicted, forced expiratory volume in one second of predicted value; FVC, forced vital capacity; FVC% predicted, forced vital capacity of predicted value; FEF 25–75, forced expiratory flow 25–75%; hs-CRP, hypersensitive C-reactive protein; SD, standard deviation.

[a]P-value by Fisher's exact test.
[b]P-value by independent sample t-test.
[c]P-value by Chi-square test.

implied increased systemic inflammation and probably decreased muscle mass played important roles in bone loss.

The association between osteoporosis and COPD is multi-factorial and could be confounding (*Romme et al., 2013*). Increasing evidence suggests an association between low BMD and systemic inflammation. This may be observed in several rheumatic diseases such as systemic lupus erythematosus, rheumatoid arthritis, and inflammatory bowel disease (*Ali et al., 2009*; *Lacativa & Farias, 2010*; *Lane, 2006*). COPD is a systemic inflammatory disease with pulmonary and extra-pulmonary manifestations. A link between COPD and extra-pulmonary comorbidities such as osteoporosis, atherosclerosis, skeletal muscle dysfunction, and anemia may be explained by systemic inflammation (*Barnes & Celli, 2009*; *Sin et al., 2006*). The disease process is believed to be initiated when lung tissues are exposed to environment irritants such as tobacco smoke or air pollutants; subsequently, stimulated epithelial cells and macrophages release inflammatory mediators, which may

**Table 7** Binary logistic regression for multivariate analysis of osteoporosis risk factors in COPD patients.

| Variables | B | SE | Odds ratio | 95% CI | | P-value |
|---|---|---|---|---|---|---|
| | | | | Lower | Upper | |
| Age (years) | 0.076 | 0.077 | 1.08 | 0.93 | 1.26 | 0.322 |
| BMI (kg/m$^2$) | −0.276 | 0.157 | 0.76 | 0.56 | 1.03 | 0.079 |
| hs-CRP ≥ 5 (mg/L) | 4.041 | 1.686 | 58.90 | 2.09 | 1548.13 | **0.017** |
| Creatinine | −4.781 | 2.233 | 0.01 | 0.00 | 0.67 | **0.032** |
| Still smoking | 1.324 | 1.015 | 3.76 | 0.51 | 27.49 | 0.192 |
| Oral or inhaled steroid >6 months | −1.570 | 1.423 | 0.21 | 0.01 | 3.39 | 0.270 |
| FEV1 | 0.458 | 1.455 | 1.58 | 0.09 | 27.37 | 0.753 |
| Constant | 3.733 | 8.196 | 41.82 | | | 0.649 |

**Notes.**

Abbreviations: BMI, body mass index; hs-CRP, hypersensitive C-reactive protein; FEV1, forced expiratory volume in one second; CI, confidence interval.
When the p-value of analysis reached statistical significance (i.e., <0.05), we showed it in bold.

directly or indirectly damage the lungs and other specific organs. In addition to the lungs, these mediators were also detected in peripheral blood (*Gan et al., 2004*).

Several inflammatory markers are associated with low BMD or increased fracture risk. This study revealed that increased hs-CRP was associated with osteoporosis development in COPD patients after adjustment for common confounding factors including age and BMI, which is in agreement with the *Rittayamai, Chuaychoo & Sriwijitkamol (2012)* study. Another study reported that TNF-α and IL-6 were independent predictors of low BMD. CRP also tended to be associated with low BMD, although statistically insignificant (*Liang & Feng, 2012*). Different case selection, for example, patients with different disease severity, might have contributed to this subtle difference. However, because tests to determine CRP level are relatively available and inexpensive (*Buess & Ludwig, 1995*), this parameter may worth considering first for detection of the existence and severity of inflammation in COPD to predict disease outcome and the possibility of extra-pulmonary comorbidity such as osteoporosis.

This study showed that increased CRP level was related to COPD severity as classified by the GOLD staging criteria and each pulmonary function parameter, and systemic steroid use, which indicated that patients with persistent systemic inflammation had an increased need to use systemic steroid to either prevent acute exacerbation or relieve the symptoms of airway obstruction. These findings were in line with several previous studies. CRP was negatively related to FEV1 (*De Torres et al., 2008*; *Saetta, 1999*), FEV1% predicted (*Saetta, 1999*), and FEV1, FVC, and arterial oxygen saturation (*Dahl et al., 2007*). Furthermore, Sin and Man found that the negative association between CRP, and FEV1 and FVC, in COPD patients was much stronger in men than in women (*Sin & Man, 2003*). These findings suggest that persistent systemic inflammation in COPD links worse lung condition with extra-pulmonary comorbidities such as osteoporosis or general cachexia and explains why some studies revealed that poor pulmonary function parameters such as FEV1 were associated with osteoporosis development (*Lin et al., 2015*; *Watanabe et al., 2015*).

Low creatinine level was associated with osteoporosis in this study. Lower creatinine level could indicate two possibilities, one was relatively better renal function, and the other was relatively lower muscle mass. Serum creatinine is a metabolite of creatine phosphate, which mostly originates from skeletal muscle with a stable breakdown rate. Therefore, serum creatinine level directly reflects the amount of muscle mass unless renal function changes (*Huh et al., 2015*; *Kim et al., 2016*). A recent study reported that low serum creatinine was related to low appendicular muscle mass and low BMD in subjects with a glomerular filtration rate >60 mL/min/1.73 m$^2$. Sarcopenia and osteoporosis should be considered in male and female patients with creatinine levels <0.88 mg/dL and <0.75 mg/dL, respectively (*Huh et al., 2015*). Loss of fat-free mass was related to COPD severity and low BMD (*Bolton et al., 2004*). In the questionnaire interivew of the present study, most COPD patients had no history of renal diseases except only one COPD patient in non-osteoporotic group was reported to have chronic kidney disease (Table 4). And since it is well known that good renal function is not related to the development of osteoporosis, the only explanation of the association between low creatinine level and osteoporosis was that low muscle mass relating to the occurrence of osteoporosis. For the findings, we suggest that the occurrence of sarcopenia and increased risk of osteoporosis should be considered in COPD patients with low creatinine levels, although normal creatinine levels should not exclude sarcopenia or chronic kidney disease because prevalences of both sarcopenia and chronic kidney disease may be increased in COPD patients (*Bolton et al., 2004*; *Chen et al., 2013*; *Incalzi et al., 2010*). In this situation, a low creatinine level would be a more sensitive indicator of sarcopenia. Because measurement of creatinine is simple and inexpensive, this parameter may be a useful screening tool for sarcopenia and osteoporosis in COPD patients.

Recently, some studies reported that increased homocysteine, decreased vitamin B12, and decreased folate levels might be related to increased fracture risk or osteoporosis, although these findings remain controversial (*Cagnacci et al., 2003*; *Fratoni & Brandi, 2015*). A study on whether homocysteine, vitamin B12, and folate play roles in osteoporosis development in COPD patients is still lacking. The present study included these items but did not find obvious associations between them and osteoporosis in male COPD patients.

There were several limitations in the present study. First, the limited case number resulted in insufficient statistical power for analysis of some variables. Second, this study enrolled stable male COPD patients; thus, the majority of patients belonged to GOLD stages II and III, which were not the group of the most severe disease with most significant inflammation. Third, certain variables that might have confounded osteoporosis development, such as vitamin D deficiency, were not included. Furthermore, assessments of calcium intake amount were based solely on frequency of milk or calcium supplement consumption, and intensity of physical activity was based on the average hours spent exercising per week or the nature of the daily employed work. Differing intensity levels of different kinds of exercise or daily activity were not considered. Thus, the precise role of calcium intake amount and intensity of physical activity were not conclusive in this study. Moreover, recall bias might have existed in the questionnaire interview. Further, this study did not include other inflammation markers such as IL-6 or TNF-α, and the CRP level was only measured once. There was no continuous monitoring to detect fluctuations in inflammation marker

level along the disease course and the effect on the pathogenesis of osteoporosis. Another limitation was the cross-sectional design of the study. Further comprehensive prospective cohort studies are needed to confirm the causal relationships and clarify the underlying mechanisms.

## CONCLUSIONS

The total prevalence of osteoporosis and osteopenia in male Taiwanese COPD patients is higher than that in age-matched male subjects and increased CRP level, which indicated systemic inflammation is an independent risk factor for osteoporosis development. Low creatinine level in COPD patients should raise the suspicion of sarcopenia and associated increased risk of osteoporosis.

## ACKNOWLEDGEMENTS

The authors are honored to acknowledge the assistance of the Chiayi County Health Bureau and local public health centers.

### Funding

This study was supported by grants CMRPG670091, CMRPG670092, MRPG670093 and CMRPG670094 provided by the Chang Gung Medical Research Fund. The funders had no role in study design, data collection and analysis, decision to publish, or preparation of the manuscript.

### Grant Disclosures

The following grant information was disclosed by the authors:
Chang Gung Medical Research Fund: CMRPG670091, CMRPG670092, MRPG670093, CMRPG670094.

### Competing Interests

The authors declare there are no competing interests.

### Author Contributions

- Chu-Hsu Lin conceived and designed the experiments, performed the experiments, wrote the paper, prepared figures and/or tables.
- Kai-Hua Chen analyzed the data, contributed reagents/materials/analysis tools, reviewed drafts of the paper.
- Chien-Min Chen analyzed the data, prepared figures and/or tables, reviewed drafts of the paper.
- Chia-Hao Chang analyzed the data, contributed reagents/materials/analysis tools.
- Tung-Jung Huang conceived and designed the experiments, reviewed drafts of the paper, communication with Chiayi County Health Bureau and local public health centers.
- Chia-Hung Lin analyzed the data.

## Human Ethics

The following information was supplied relating to ethical approvals (i.e., approving body and any reference numbers):

Written informed consent was obtained from all participants, and the study was approved by the local Institutional Review Board of Chiayi Chang Gung Memorial Hospital (IRB number: 96-0495B).

## Data Availability

The raw data has been provided as Supplemental Information 1.

## Supplemental Information

Supplemental information for this article can be found online at http://dx.doi.org/10.7717/peerj.4232#supplemental-information.

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
