# Peer review of "Risk factors for osteoporosis in male patients with chronic obstructive pulmonary disease in Taiwan"

_PeerJ, doi:10.7717/peerj.4232_

## Round 0.1 · original submission · Major Revisions

Thank you for your submission and your interesting study. As requested by the reviewers, before I can give a more thorough review, you will need to upload the data in an Excel document as not all reviewers have access to SPSS. In terms of the remaining comments, please read, review, and respond to each of the comments by the reviewers.

I look forward to reviewing your resubmission.

Scotty

Reviewer 1 ·

Basic reporting

no comment

Experimental design

1) In Table 1, basic characteristics of controls should also be presented for comparison with COPD patients. In order to conclude that osteoporosis is more prevalent in COPD patients, data should be adjusted for various confounders including smoking status, renal function, the presence or absence of diabetes, and so on. It is also important for the authors to exclude COPD in control subjects by spirometric test.
2) According to the authors, controls included family members of COPD patients. But exactly how did the authors recruit these subjects? And how many were family members of COPD patients?

Validity of the findings

3) In Table 2, mean values of CRP and proportion of subjects with CRP higher than 5.0 should be shown in each group.
4) In Table 2, eGFR should be included. If the authors considered creatinine as an indicator of muscle mass, how could they be sure that renal function was the same?
5) And in Table 5, important factors missing here, such as systemic steroid use, current smoking, respiratory function and so on, should also be included. Whether or not CRP contribution to osteoporosis was dependent on respiratory dysfunction is particularly important, because higher CRP is apparently associated with impaired respiratory function in Table 4.
6) In Table 4 smoking status in high CRP group (1/8, 1/8, 1/8?) seems wrong and should be corrected. And correct information should be used to analyze determinants of osteoporosis (See comment 4)).
7) Serum levels of 25-hydroxyvitamin D should be measured. Vitamin D deficiency is a well-known risk factor of COPD-associated osteoporosis.

Additional comments

COPD-associated osteoporosis is common, but its precise prevalence as well as its mechanism remains largely unknown. Thus, the subject is important. The manuscript is generally well written. However, some critical information is lacking, and statistical analysis seems incomplete. Comparison between control and COPD subjects is far from conclusive because of the small sample size and the way of subject recruitment.

Reviewer 2 ·

Basic reporting

Clear, unambiguous, professional English language used throughout - yes

Discussion - Findings very well discussed
Specific comments
Lines
183-188
“However, the prevalence of osteoporosis in our study was lower than the 40% reported in a recent study also conducted in Chiayi, Taiwan (Lin 185 et al., 2015). This may be explained by differences in subject enrollment and osteoporosis. That study enrolled male and female COPD patients and defined osteoporosis as a BMD T-score <−2.5 or the presence of thoracolumbar vertebral compression fracture on radiography.” → Very good to put the difference in prevalence in perspective.
Financial support and sponsorship, conflicts of interest and acknowledgments indicated.

Intro & background to show context. Literature well referenced & relevant
General comments
Reasons for study well argued; referencing correct and up-to-date
Specific comments
Lines
50-52
“It affects >5% of the population (Centers for Disease Control and Prevention, 2012) and is predicted to become the third leading cause of death in the world by 2020 (Murray & Lopez, 1997).” → Which population, world-wide?
72-74
“Therefore, this study primarily aimed to investigate risk factors for osteoporosis, particularly whether increased CRP is a predictor for osteoporosis development in male Taiwanese COPD patients.” → Why did you specifically choose “high-sensitivity CRP”? Better predicator of osteoporosis than IL-6?

Structure conforms to PeerJ standards, discipline norm → yes

Figures and relevant, high quality, well labeled & described
Specific comments
Table 2
a, b, and c (indicating the different statistical tests) are confusing, one expects that they would indicate significance → Suggest that the description (or indication) of what test was used for which analysis being put in the “Statistical analysis” section – lines 133-135, and significance being highlighted in the table.
Table 3
See comment on Table 2
Table 4
See comment on Table 2
Are these still the COPD patients?

Raw data supplied, but as I don’t use SPSS, I can’t read the file. Can you please provide the data in an Excel file?

Experimental design

Original primary research within Scope of the journal → yes

Research question well defined, relevant & meaningful. It is stated how the research fills an identified knowledge gap → yes

Rigorous investigation performed to a high technical & ethical standard
General comments
• In- and exclusion criteria provided
• Written informed consent
o Possible to see an English version of the consent form?
• IRB# provided
Specific comments
Lines
79
“It enrolled male COPD patients aged ≥45 years from……”
AND
83-85
“Subjects who were willing to undergo anthropometric measurements ….” → How and by whom were the COPD patients recruited?
93-107
Description of the “Questionnaire interview and medical records review” → Would recommend putting the information in a table – would make reading it much easier.
120-121
“Lung examination: Chest radiography and spirometry data were interpreted by a chest physician.” → Not possible to get more than one chest physician to interpret the CxR? Was he/she a radiologist?

Methods described with sufficient detail & information to replicate → yes

Validity of the findings

Validity of findings
Rationale & benefit to literature is clearly stated

Data is robust, statistically sound, & controlled
Data analysis – correct
Results – well described
Specific comments
Lines
161-163
“The high hs-CRP group tended to have lower BMI, although there were no significant differences in age, smoking habits, and history of acute exacerbation (Table 4).” → Suggest moving this up to the previous paragraph – keep the hs-CRP information together.
159-160
“CRP showed strong associations with COPD severity and pulmonary function parameters (including COPD GOLD stage, FEV1, FEV1% predicted, FVC, FVC% predicted, FEV1/FVC, and FEF 25–75%) and steroid use.” → Can you differentiate the significance according to the different GOLD stages?

Additional comments

Conclusions are well stated, linked to original research question & limited to supporting results → yes

General comments → Very well written manuscript

---

## Round 0.2 · accepted · Accept

Thank you for your attention to the editorial and reviewer's comments and concerns. The manuscript is now acceptable in its current form. Congratulations!

Staff note: we confirm that the one remaining concern from Reviewer 1 ("Comment from reviewer: I don’t see the clarification.") was actually included in the manuscript (line 67-69 of the reviewing PDF)

Reviewer 1 ·

Basic reporting

good

Experimental design

no comment

Validity of the findings

Conclusions are now well stated and supported by the results.

Additional comments

Revision has been well done.

Reviewer 2 ·

Basic reporting

Lines 50-52
“It affects >5% of the population (Centers for Disease Control and Prevention, 2012) and is predicted to become the third leading cause of death in the world by 2020 (Murray & Lopez, 1997).” → Which population, world-wide?
Reply:
Much thanks for the reviewer’s carefully reading. The population meant the people in the United States. It seemed not suitable to put this statement here since it was not world representative. We have deleted it in the text and in the reference. (Page 4 # line 51-2; Page 20 # line 343-4)
Comment from reviewer: Addressed√

Lines 72-74
“Therefore, this study primarily aimed to investigate risk factors for osteoporosis, particularly whether increased CRP is a predictor for osteoporosis development in male Taiwanese COPD patients.” → Why did you specifically choose “high-sensitivity CRP”? Better predicator of osteoporosis than IL-6?
Reply:
We have chosen CRP because it was a common, cheap and quite available test in our hospital. To clarify this, we have addressed it in Page 5 # line 75-6.
Comment from reviewer: I don’t see the clarification.

Experimental design

Raw data supplied, but as I don’t use SPSS, I can’t read the file. Can you please provide the data in an Excel file?
Reply:
OK, we will upload the raw data in Excel file.
Comment from reviewer: Addressed√.

The study enrolled Taiwanese people, therefore, there was no English version of informed consent. We have translated the inform consent form into English for the reviewer.
Comment from reviewer: Addressed√.

Lines 79
“It enrolled male COPD patients aged ≥45 years from……”
AND
Lines 83-85
“Subjects who were willing to undergo anthropometric measurements ….” → How and by whom were the COPD patients recruited?
Reply:
I ask my several colleagues, who are also chest physicians to help us to enroll the out-patient COPD patients in chest clinic in Chiayi Chang Gung Memorial Hospital. If the patient had the interests to know more about the project, the chest physicians including me, would call the research assistant to help to explain the project more completely. If the patients agreed to participate the project, they would be asked to read and sign the inform consents.
Comment from reviewer: Addressed√.

Lines 120-121
“Lung examination: Chest radiography and spirometry data were interpreted by a chest physician.” → Not possible to get more than one chest physician to interpret the CxR? Was he/she a radiologist?
Reply:
Chest radiography and spirometry data were interpreted only by me. I am a chest physician but not a radiologist. However, in Taiwan, interpretation of chest X-ray is essential for becoming a qualified chest physician.
Comment from reviewer: Addressed√.

Validity of the findings

Tables
Table 1 Looking good
Table 2 Control (36) COPD (59)
L-spine BMD (35) L-spine BMD (58)
Hip BMD (57)
Chronic hepatitis (58)
If you weren’t able to measure these, mention that at the bottom of the table
Table 3 Variables: “B”?
Table 4 hs-CRP – put in either the mean ± SD or the %, it doesn’t read well with both in
Table 5 Looking good
Table 6 Looking good
Table 7 Variables: “B”?

Specific comments
Table 2
a, b, and c (indicating the different statistical tests) are confusing, one expects that they would indicate significance → Suggest that the description (or indication) of what test was used for which analysis being put in the “Statistical analysis” section – lines 133-135, and significance being highlighted in the table.
Reply:
We have highlighted all significant value in bold in all tables according to the reviewer’s suggestion.
We have explained the rules of statistical analysis in the “Data Analysis” section (Page 9 # line 135-42).. But we think the analysis process is complicated. It is very difficult to descript it without such labeling in each table. Therefore, we still kept them in each table and we think the reader would not be confused after highlighting each significant value in “bold”.
Table 3
See comment on Table 2
Reply:
The same reply as in Table 2.
Table 4
See comment on Table 2
Are these still the COPD patients?
Reply:
The same reply as in Table 2. Table 4 was the comparison of “COPD patients” with low or high CRP. To avoid be confused by the reader, we have added the word “COPD” in the title of the table. (see Table 5, changed from Table 4)
Comment from reviewer: Addressed√.

Lines 159-160
“CRP showed strong associations with COPD severity and pulmonary function parameters (including COPD GOLD stage, FEV1, FEV1% predicted, FVC, FVC% predicted, FEV1/FVC, and FEF 25–75%) and steroid use.” → Can you differentiate the significance according to the different GOLD stages?
Reply:
Because the case number in this study is small. Further analysis with cases divided into GOLD stage I-IV to compare the pulmonary function parameters difference between low and high hs-CRP revealed loss of most of the significance possibly due to insufficient statistical power. However, the tendency towards lower pulmonary function parameters associated with increased hs-CRP seemed still existed. The results were as following (Tables provided).
Comment from reviewer: Addressed√.

Additional comments

Lines 161-163
“The high hs-CRP group tended to have lower BMI, although there were no significant differences in age, smoking habits, and history of acute exacerbation (Table 4).” → Suggest moving this up to the previous paragraph – keep the hs-CRP information together.
Reply:
We have moved it up to the previous paragraph according to the reviewer’s suggestions. (Page 11 # line 169-73)
Comment from reviewer: Addressed√.